# SURROGATE-BASED CONSTRAINED LANGEVIN SAMPLING WITH APPLICATIONS TO OPTIMAL MATERIAL CONFIGURATION DESIGN

## ABSTRACT

We consider the problem of generating configurations that satisfy physical constraints for optimal material nano-pattern design, where multiple (and often conflicting) properties need to be simultaneously satisfied. Consider, for example, the trade-off between thermal resistance, electrical conductivity, and mechanical stability needed to design a nano-porous template with optimal thermoelectric efficiency. To that end, we leverage the posterior regularization framework and show that this constraint satisfaction problem can be formulated as sampling from a Gibbs distribution. The main challenges come from the black-box nature of those physical constraints, since they are obtained via solving highly non-linear PDEs. To overcome those difficulties, we introduce Surrogate-based Constrained Langevin dynamics for black-box sampling. We explore two surrogate approaches. The first approach exploits zero-order approximation of gradients in the Langevin Sampling and we refer to it as Zero-Order Langevin. In practice, this approach can be prohibitive since we still need to often query the expensive PDE solvers. The second approach approximates the gradients in the Langevin dynamics with deep neural networks, allowing us an efficient sampling strategy using the surrogate model. We prove the convergence of those two approaches when the target distribution is $\log$-concave and smooth. We show the effectiveness of both approaches in designing optimal nano-porous material configurations, where the goal is to produce nano-pattern templates with low thermal conductivity and reasonable mechanical stability.

## 1 INTRODUCTION

In many real-world design problems, the optimal design needs to simultaneously satisfy multiple constraints, which can be expensive to estimate. For example, in computational material design, the goal is to come up with material configurations, or *samples*, satisfying a list of physical constraints that are given by black-box numerical Partial Differential Equations (PDE) solvers. Such solvers (for example, the Boltzmann Transport Equation solver) are often complex, expensive to evaluate, and offer no access to their inner variables or their gradients.

We pose this design-under-constraints problem as sampling from a Gibbs distribution defined on some compact support. The problem of sampling from a distribution with unknown likelihood that can only be point-wise evaluated is called black-box sampling (Chen & Schmeiser, 1998; Neal, 2003). We show in this paper that constrained black-box sampling can be cast as a *constrained* Langevin dynamics with gradient-free methods. Zero-order optimization via Gaussian smoothing was introduced in Nesterov & Spokoiny (2017) and extended to black-box sampling with Langevin dynamics in Shen et al. (2019). We extend this approach to the constrained setting from a black-box density with compact support.

However, one shortcoming of this approach is that it is computationally very expensive since it requires repeatedly querying PDE solvers in order to get an estimate of the gradient. To alleviate computational issues, we propose *Surrogate Model Based Langevin dynamics*, that consists of two steps: (i) Learning (using training data) an approximation of the *gradient* of the potential of the Gibbs distribution. We show that learning the gradient, rather than the potential itself, is important for the

mixing of the Langevin dynamics towards the target Gibbs distribution. We devise several objective functions, as well as deep neural-network architectures for parameterizing the approximating function class, for learning the gradient of the potential function. (ii) We then use the surrogate gradient model in the constrained Langevin dynamics *in lieu of* the black-box potential. Using the surrogate enables more efficient sampling, since it avoids querying the expensive PDE solvers, and obtaining gradients is as efficient as evaluating the functions themselves using automatic differentiation frameworks such as PyTorch or TensorFlow.

To summarize, our main contributions are as follows:

1. We cast the problem of generating samples under constraints in the black-box setting as sampling from a Gibbs distribution.

2. We introduce Constrained Zero-Order Langevin Monte Carlo, using projection or proximal methods, and provide the proof of its convergence to the target Gibbs distribution.

3. We introduce Surrogate Model Based Projected Langevin Monte Carlo via learning the gradient of the potential of the Gibbs distribution using deep neural networks or reproducing kernel spaces, and prove its convergence to the target distribution when used in conjunction with projection or proximal based methods. We shed the light on the importance of the approximation of the gradient of the potential, and we show how to achieve this using Hermite and Taylor learning.

4. We showcase the usability and effectiveness of the proposed methods for the design of nano-porous configurations with improved thermoelectric efficiency. The design consists of finding new configurations with optimized pore locations, such that the resulting configurations have favorable thermal conductivity (i.e., minimal $\kappa$) and desired mechanical stability (von Mises Stress $\sigma \leq \tau$, where $\tau$ is some preset threshold). [1]

## 2 FROM CONSTRAINTS SATISFACTION TO SAMPLING FROM A GIBBS DISTRIBUTION: POSTERIOR REGULARIZATION

In black-box optimization problems (such as the material design under consideration), the goal is to find a posterior distribution $q$ of samples satisfying a list of equality and inequality constraints: $\psi_j(x) = y_k, j = 1 \dots C_e$, and $\phi_k(x) \leq b_k, k = 1 \dots C_i$ where $x \in \Omega$ and $\Omega \subset \mathbb{R}^d$ is a bounded domain. We assume a prior distribution $p_0$ (whose analytical form is known). The main challenge in black-box optimization is that the functions $\psi_j$ and $\phi_k$ can be only evaluated point-wise, and neither do we have functional forms nor access to their gradients. For example, $\psi$ and $\phi$ might be obtained via aggregating some statistics on the solution of a nonlinear PDE given by a complex solver.

To make the problem of learning under constraints tractable, we choose Lagrangian parameters $\lambda_j > 0$ and obtain the following relaxed objective:

$$\min_{q, \int_\Omega q(x)=1} \text{KL}(q, p_0) + \sum_{j=1}^{C_e} \lambda_j \mathbb{E}_{x \sim q}(\psi_j(x) - y_k)^2 + \sum_{k=1}^{C_i} \lambda_k \mathbb{E}_{x \sim q}(\phi_k(x) - b_k)_+ \tag{1}$$

The formulation in Eq. 1 is similar in spirit to the posterior regularization framework of Ganchev et al. (2010); Hu et al. (2018). However, we highlight two differences: (i) our focus is on constrained settings (where $\Omega$ is bounded), and (ii) we assume a black-box setting. We first obtain:

**Lemma 1** (Constraint Satisfaction as Sampling from a Gibbs Distribution). *The solution to the distribution learning problem given in Eq. 1 is given by:*

$$\pi(x) = \frac{\exp(-U(x))}{Z} \mathbb{1}_{x \in \Omega} \tag{2}$$

*where $U(x) = -\log p_0(x) + \sum_{j=1}^{C_e} \lambda_j (\psi_j(x) - y_k)^2 + \sum_{k=1}^{C_i} \lambda_k (\phi_k(x) - b_k)_+$ and $Z = \int_{x \in \Omega} \exp(-U(x)) \, dx$.*

Lemma 1 shows that the constraint satisfaction problem formulated in Eq. 1 amounts to sampling from a Gibbs distribution defined on a compact support given in Eq. 2. Sampling from a Gibbs

---

[1]Note that both properties $\kappa$ and $\sigma$ for a given configuration are obtained by numerically solving highly non-linear PDEs. The material configuration is defined by the pore locations, the material used, and the response of the material to heat (thermal) or stress (mechanical) flows.

distribution (also known as Boltzmann distribution) has a long history using Langevin dynamics. In the white-box setting when the functions defining the constraints have explicit analytical forms as well as their gradients, Langevin dynamics for Gibbs distribution sampling defined on a compact domain $\Omega$ and their mixing properties were actively studied in Bubeck et al. (2015); Brosse et al. (2017). In the next Section, we provide a more detailed review.

**Remark 1** (Relation to Bayesian Optimization). *While in Bayesian optimization we are interested in finding a point that satisfies the constraints, in our setting we are interested in finding a distribution of candidate samples that satisfy (black-box) constraints. See (Suzuki et al., 2019) for more details.*

**Remark 2.** *For the rest of the paper, we will assume $p_0$ to be the uniform distribution on $\Omega$, which means that its gradients are zero on the support of the domain $\Omega$. Otherwise, if $p_0$ is known and belongs to, for instance, an exponential family or a generative model prior (such as normalizing flows), we can sample from $\pi$ using a mixture of black-box sampling on the constraints ($\psi_j, \phi_k$) and white-box sampling on $\log(p_0)$.*

## 3 WHITE-BOX SAMPLING: CONSTRAINED LANGEVIN DYNAMICS

We review in this section Langevin dynamics in the unconstrained case ($\Omega = \mathbb{R}^d$) and the constrained setting ($\Omega \subset \mathbb{R}^d$). Below, $\|\cdot\|$ denotes the Euclidean norm unless otherwise specified. We are interested in sampling from

$$\pi(x) = \frac{1}{Z} \exp(-U(x)) \mathbb{1}_{x \in \Omega}, \tag{3}$$

**Preliminaries.** We give here assumptions, definitions and few preliminary known facts that will be useful later. Those assumptions are commonly used in Langevin sampling analysis (Dalalyan, 2017; Bubeck et al., 2015; Brosse et al., 2017; Durmus et al., 2019).

1. **Assumption A:** We assume $\Omega$ is a convex such that $0 \in \Omega$, $\Omega$ contains a Euclidean ball of radius $r$, and $\Omega$ is contained in a Euclidean ball of radius R. (For example, $\Omega$ might encode box constraints.) The projection onto $\Omega$, $P_\Omega(x)$ is defined as follows: for all $x \in \Omega$, $P_\Omega(x) = \arg\min_{z \in \Omega} \|x - z\|^2$. Let $R = \sup_{x,x' \in \Omega} \|x - x'\| < \infty$.

2. **Assumption B:** We assume that $U$ is convex, $\beta$-smooth, and with bounded gradients:

$$\|\nabla_x U(x) - \nabla_y U(y)\| \leq \beta \|x - y\|, \quad \forall x, y \in \Omega \text{ ($\beta$-smoothness).}$$

$$\|\nabla U(x)\| \leq L, \quad \forall x \in \Omega \text{ (Boundedness).}$$

The Total Variation (TV) distance between two measures $\mu, \nu$ is defined as follows: $TV(\mu, \nu) = \sup_A |\mu(A) - \nu(A)|$. Pinsker Inequality relates KL divergence to TV: $TV(\mu, \nu) \leq \sqrt{2\text{KL}(\mu, \nu)}$.

**Unconstrained Langevin Dynamics.** In the unconstrained case, the goal is to sample from a Gibbs distribution $\pi(x) = \exp(-U(x))/Z$ that has unbounded support. This sampling can be done via the Langevin Monte Carlo (LMC) algorithm, which is given by the following iteration:

$$X_{k+1} = X_k - \eta \nabla_x U(X_k) + \sqrt{2\lambda\eta}\xi_k, \quad k = 0 \ldots K - 1 \text{ (LMC)}, \tag{4}$$

where $\xi_k \sim \mathcal{N}(0, I_d)$, $\eta$ is the learning rate, and $\lambda > 0$ is a variance term.

**Constrained Langevin Dynamics.** In the constrained case, the goal is to sample from $\pi(x) = \exp(-U(x))/Z \mathbb{1}_{x \in \Omega}$. We discuss two variants:

***Projected Langevin Dynamics.*** Similar to projected gradient descent, Bubeck et al. (2015) introduced Projected Langevin Monte Carlo (PLMC) and proved its mixing propreties towards the stationary distribution $\pi$. PLMC is given by the following iteration :

$$X_{k+1} = P_\Omega \left( X_k - \eta \nabla_x U(X_k) + \sqrt{2\lambda\eta}\xi_k \right), \quad k = 0 \ldots K - 1 \text{ (PLMC)}, \tag{5}$$

In essence, PLMC consists of a single iteration of LMC, followed by a projection on the set $\Omega$ using the operator $P_\Omega$.

***Proximal Langevin Dynamics.*** Similar to proximal methods in constrained optimization, Brosse et al. (2017) introduced Proximal LMC (ProxLMC) that uses the iteration:

$$X_{k+1} = \left(1 - \frac{\eta}{\gamma}\right) X_k - \eta \nabla_x U(X_k) + \frac{\eta}{\gamma} P_\Omega(X_k) + \sqrt{2\lambda\eta}\xi_k, k = 0 \dots K - 1, \quad \textbf{(ProxLMC)} \quad (6)$$

where $\eta$ is the step size and $\gamma$ is a regularization parameter. In essence, ProxLMC (Brosse et al., 2017) performs an ordinary LMC on $U^\gamma(x) = U(x) + i_\Omega^\gamma(x)$, where $i_\Omega^\gamma(x)$ is the proximal operator:

$$i_\Omega^\gamma(x) = \inf_y i_\Omega(x) + (2\gamma)^{-1} \|x - y\|^2 = (2\gamma)^{-1} \|x - P_\Omega(x)\|^2,$$

where $i_\Omega(x) = 0$ for $x \in \Omega$ and $i_\Omega(x) = \infty$ for $x \notin \Omega$. Therefore, the update in Eq. 6 is a regular Langevin update (as in Eq. 4) with potential gradient $\nabla_x U^\gamma(x) = \nabla_x U(x) + \gamma^{-1}(x - P_\Omega(x))$.

We denote by $\mu_K^{\text{PLMC}}$ and $\mu_K^{\text{ProxLMC}}$ the distributions of $X_K$ obtained by iterating Eq. 5 and Eq. 6 respectively. Under Assumptions **A** and **B**, both these distributions converge to the target Gibbs distribution $\pi$ in the total variation distance. In particular, Bubeck et al. (2015) showed that for $\eta = \tilde{\Theta}(R^2/K)$, we obtain:

$$TV(\mu_K^{\text{PLMC}}, \pi) \leq \varepsilon \text{ for } K = \tilde{\Omega}(\varepsilon^{-12}d^{12}). \quad (7)$$

Likewise, Brosse et al. (2017) showed that for $0 < \eta \leq \gamma(1 + \beta^2\gamma^2)^{-1}$, we obtain:

$$TV(\mu_K^{\text{ProxLMC}}, \pi) \leq \varepsilon \text{ for } K = \tilde{\Omega}(\varepsilon^{-6}d^5), \quad (8)$$

where the notation $\alpha_n = \tilde{\Omega}(\beta_n)$ means that there exists $c \in \mathbb{R}, C > 0$ such that $\alpha_n \geq C\beta_n \log^c(\beta_n)$.

## 4 CONSTRAINED LANGEVIN DYNAMICS IN THE BLACK-BOX SETTING

We now introduce our variants of constrained LMC for the black-box setting where explicit potential gradients are unavailable. We explore in this paper two strategies for approximating the gradient of $U$ in the black-box setting. In the first strategy, we borrow ideas from derivative-free optimization (in particular, evolutionary search). In the second strategy we learn a *surrogate deep model* that approximates the gradient of the potential. Below, let $G : \Omega \to \mathbb{R}^d$ be a vector valued function that approximates the gradient of the potential, $\nabla_x U$. We make:

**Assumption C.** The surrogate gradient $G$ satisfies $\mathbb{E} \|G(Y_k)\|^2 < \infty, \forall k$.

**Surrogate Projected Langevin Dynamics.** Given $Y_0$, the Surrogate Projected LMC (S-PLMC) replaces the potential gradient $\nabla_x U$ in Eq. 5 with the surrogate gradient $G$:

$$Y_{k+1} = P_\Omega \left(Y_k - \eta \mathbf{G}(Y_k) + \sqrt{2\lambda\eta}\xi_k\right), k = 0 \dots K - 1 \quad \textbf{(S-PLMC)} \quad (9)$$

**Surrogate Proximal Langevin Dynamics.** Similarly, the Surrogate Proximal LMC (S-ProxLMC) replaces the unknown potential gradient $\nabla_x U$ in Eq. 6 with the gradient surrogate $G$:

$$Y_{k+1} = \left(1 - \frac{\eta}{\gamma}\right) Y_k - \eta \mathbf{G}(Y_k) + \frac{\eta}{\gamma} P_\Omega(Y_k) + \sqrt{2\lambda\eta}\xi_k, k = 0 \dots K - 1 \quad \textbf{(S-ProxLMC)} \quad (10)$$

We now present our main theorems on the approximation properties of surrogate LMC (S-PLMC, and S-ProxLMC). We do so by bounding the total variation distance between the trajectories of the surrogate Langevin dynamics (S-PLMC, and S-ProxLMC) and the true LMC dynamics (PLMC and ProxLMC). Theorem 1 is an application of techniques in Stochastic Differential Equations (SDE) introduced in Dalalyan & Tsybakov (2012) and is mainly based on a variant of Grisanov's Theorem for change of measures (Lipster & Shiryaev, 2001) and Pinsker's Inequality that bounds total variation in terms of Kullback-Leibler divergence.

**Theorem 1** (S-PLMC and S-ProxLMC Mixing Properties). *Under Assumption **C**, we have:*

1. ***S-PLMC Convergence.*** *Let $\mu_K^{\text{PLMC}}$ be the distribution of the random variable $X_K$ obtained by iterating **PLMC** Eq. 5, and $\mu_K^{\text{S-PLMC}}$ be the distribution of the random variable $Y_K$ obtained by iteration **S-PLMC** given in Eq. 9. We have:*

$$TV(\mu_K^{\text{S-PLMC}}, \mu_K^{\text{PLMC}}) \leq \sqrt{\frac{\eta}{\lambda} \left(\sum_{k=0}^{K-1} \mathbb{E} \|G(Y_k) - \nabla_x U(Y_k)\|^2 + K\beta^2 R^2\right)^{\frac{1}{2}}}. \quad (11)$$

2. **S-ProxLMC Convergence.** *Let $\mu_K^{\text{ProxLMC}}$ be the distribution of the random variable $X_K$ obtained by iterating **ProxLMC** Eq. 6, and $\mu_K^{\text{S-ProxLMC}}$ be the distribution of the random variable $Y_K$ obtained by iterating **S-ProxLMC** given in Eq. 10. We have:*

$$TV(\mu_K^{\text{S-ProxLMC}}, \mu_K^{\text{ProxLMC}}) \leq \sqrt{\frac{\eta}{2\lambda}} \left( \sum_{k=0}^{K-1} \mathbb{E} \left\| G(X_k) - \nabla_x U(X_k) \right\|^2 \right)^{\frac{1}{2}}. \tag{12}$$

From Theorem 1, we see that it suffices to approximate the potential gradient $\nabla_x U(X)$ (and not the potential $U(X)$) in order to guarantee convergence of surrogate-based Langevin sampling. Using the triangle inequality, and combining Theorem 1 and bounds in Eqs 7 and 8 we obtain:

**Theorem 2.** *(Convergence of Surrogate Constrained LMC to the Gibbs distribution.) Under assumptions A,B and C we have:*

1. *Assume in S-PLMC that there exists $\delta > 0$ such that $\mathbb{E} \left\| G(Y_k) - \nabla_x U(Y_k) \right\|^2 \leq \delta, \forall k \geq 0$. Set $\lambda = 1$, and $\eta = \tilde{\Theta}(\min(R^2/K, \alpha/K^2))$ where $\alpha = 1/(\delta + \beta^2 R^2)$. Then for $K = \tilde{\Omega}(\varepsilon^{-12}d^{12})$, we have:*

$$TV(\mu_K^{\text{S-PLMC}}, \pi) \leq \varepsilon.$$

2. *Assume in S-ProxLMC that there exists $\delta > 0$ such that $\mathbb{E} \left\| G(X_k) - \nabla_x U(X_k) \right\|^2 \leq \delta, \forall k \geq 0$. Set $\lambda = 1$, and $\eta = \min(\gamma(1 + \beta^2\gamma^2)^{-1}, \frac{1}{\delta K^2})$. Then for $K = \tilde{\Omega}(\varepsilon^{-6}d^5)$ we have:*

$$TV(\mu_K^{\text{S-ProxLMC}}, \pi) \leq \varepsilon.$$

## 5 ZERO-ORDER CONSTRAINED LANGEVIN DYNAMICS

In zero-order optimization (Nesterov & Spokoiny, 2017; Duchi et al., 2015; Ghadimi & Lan, 2013; Shen et al., 2019), one considers the Gaussian smoothed potential $U_\nu$ defined as $U_\nu(x) = \mathbb{E}_{g \sim \mathcal{N}(0, I_d)} U(x + \nu g)$, and its gradient is given by $\nabla_x U_\nu(x) = \mathbb{E}_g \frac{U(x + \nu g) - U(x)}{\nu} g$. The following is a Monte Carlo estimate of $\nabla_x U_\nu(x)$:

$$\hat{G}_n U(x) = \frac{1}{n} \sum_{j=1}^{n} \left( \frac{U(x + \nu g_j) - U(x)}{\nu} \right) g_j, \tag{13}$$

where $g_1, \ldots g_n$ are i.i.d. standard normal vectors.

Zero-Order sampling from $\log$-concave densities was recently studied in Shen et al. (2019). We extend it here to the constrained sampling case of $\log$-concave densities with compact support. We define Constrained Zero-Order Projected LMC (Z-PLMC) and Zero-Order Proximal LMC (Z-ProxLMC) by setting $\mathbf{G(x)} = \hat{\mathbf{G}}_{\mathbf{n}}\mathbf{U(x)}$ in Eq. 9 and Eq. 10 respectively.

**Lemma 2** (Zero-Order Gradient Approximation(Nesterov & Spokoiny, 2017; Shen et al., 2019)). *Under Assumption B, we have for all $x \in \Omega$:*

$$\mathbb{E}_{g_1, \ldots, g_n} \left\| \hat{G}_n U(x) - \nabla_x U(x) \right\|^2 \leq \left( \beta\nu(d + 2)^{3/2} + (d + 1)^{\frac{1}{2}} L \right)^2 / n \tag{14}$$

Thanks to Lemma 2 that ensures uniform approximation of gradients in expectation, we can apply Theorem 2 and get the following corollary for Z-PLMC and Z-ProxLMC:

**Corollary 1** (Zero-order Constrained Langevin approximates the Gibbs distribution). *Under Assumptions A and B, let $\delta \in [0, 1]$, for $n \geq \left( \beta\nu(d + 2)^{3/2} + (d + 1)^{\frac{1}{2}} L \right)^2 / \delta$, we have the following bounds in expectation:*

1. *Set $\lambda = 1$, and $\eta = \tilde{\Theta}(\min(R^2/K, \alpha/K^2))$ where $\alpha = 1/(\delta + \beta^2 R^2)$. For $K = \tilde{\Omega}(\varepsilon^{-12}d^{12})$, we have:*

$$\mathbb{E}_{g_1, \ldots g_n} TV(\mu_K^{\text{Z-PLMC}}, \pi) \leq \varepsilon. \tag{15}$$

2. *Set $\lambda = 1$, and $\eta = \min(\gamma(1 + \beta^2\gamma^2)^{-1}, \frac{1}{\delta K^2})$. For $K = \tilde{\Omega}(\varepsilon^{-6}d^5)$ we have:*

$$\mathbb{E}_{g_1, \ldots g_n} TV(\mu_K^{\text{Z-ProxLMC}}, \pi) \leq \varepsilon. \tag{16}$$

**Remark 3.** *For simplicity, we state the above bound in terms of expectations over the randomness in estimating the gradients. It is possible to get finite-sample bounds using the Vector Bernstein concentration inequality, coupled with covering number estimates of $\Omega$ but omit them due to space.*

## 6 Surrogate Model Based Constrained Langevin Dynamics

Despite its theoretical guarantees, zero-order constrained Langevin (Z-PLMC and Z-ProxLMC) has a prohibitive computation cost as it needs $O(nK)$ black-box queries (in our case, invocations of a nonlinear PDE solver). To alleviate this issue, we introduce in this Section a *neural* surrogate model as an alternative to the gradient of the true potential.

### 6.1 Hermite Learning of Gradients: Jacobian Matching of Zero-Oder Estimates

From Theorem 2, we saw that in order to guarantee the convergence of constrained Langevin dynamics, we need a good estimate of the gradient of the potential of the Gibbs distribution. Recall that the potential given in Lemma 1 depends on $\psi_j$ and $\phi_k$, which are scalar outputs of computationally heavy PDE solvers in our material design problem. To avoid this, we propose to train surrogate neural network models approximating each PDE output *and* their gradients. Concretely, suppose we are given a training set $S$ for a PDE solver for the property $\psi$ (dropping the index $j$ for simplicity):

$$S = \{(x_i, y_i = \psi(x_i), \tilde{y}_i = \hat{G}_n \psi(x_i)), x_i \sim \rho_\Omega \text{i.i.d.}, i = 1, \ldots, N\},$$

where $\rho_\Omega$ is the training distribution and $\hat{G}_n \psi(.)$ is the zero-order estimate of the gradient of $\psi$ given in Eq. 13. We propose to learn a surrogate model belonging to a function class $\mathscr{H}_\theta$, $\hat{f}_\theta \in \mathscr{H}_\theta$, that regresses the value of $\psi$ and matches the zero-order gradient estimates as follows:

$$\min_{f_\theta \in \mathscr{H}_\theta} \frac{1}{N} \sum_{i=1}^N \{(y_i - f_\theta(x_i))^2 + \|\nabla_x f_\theta(x_i) - \tilde{y}_i\|^2\} \text{ (Z-Hermite Learning)} \qquad (17)$$

The problem in Eq. 17 was introduced and analyzed in Shi et al. (2010) where $\mathscr{H}_\theta$ is a ball in a Reproducing Kernel Hilbert Space (RKHS). Following Shi et al. (2010), we refer to this type of learning as *Hermite Learning*. In the deep learning community, this type of learning is called Jacobian matching and was introduced in Srinivas & Fleuret (2018); Czarnecki et al. (2017) where $\mathscr{H}_\theta$ is a deep neural network parameterized with weights $\theta$. When $f_\theta$ is a deep network, we can optimize this objective efficiently using common deep learning frameworks (PyTorch, TensorFlow).

(Shi et al., 2010) have shown that when $\mathscr{H}_\theta$ is an RKHS ball and when $\tilde{y}_i = \nabla_x \psi(x_i)$ are exact gradients, for a sufficiently large training set with $N = O(1/\epsilon^{1/(2r\zeta)})$ (where $r, \zeta$ are exponents in $[0, 1]$ that depend on the regularity of the function $\psi$). Under the assumption that $\psi \in \mathscr{H}_\theta$ we have: $\int_\Omega \|\nabla_x f_\theta(x) - \nabla_x \psi(x)\|^2 \rho_\Omega(x) dx \leq \epsilon$. Since we are using inexact zero-order gradients, we will incur an additional numerical error that is also bounded as shown in Lemma 2.

### 6.2 Taylor Learning of Gradients

While Jacobian matching of zero-order gradients is a sound approach, it remains expensive to construct the dataset, as we need for each point to have $2n + 1$ queries of the PDE solver. We exploit in this section the Taylor learning framework of gradients that was introduced in Mukherjee & Zhou (2006); Mukherjee & Wu (2006), and Wu et al. (2010). In a nutshell, Mukherjee & Zhou (2006) suggests to learn a surrogate potential $f_\theta$ and gradient $G_\Lambda$ that are consistent with the first-order taylor expansion. Given a training set $S = \{(x_i, y_i = \psi(x_i)), x \sim \rho_\Omega, i = 1 \ldots N\}$, Wu et al. (2010) suggest the following objective:

$$\min_{f_\theta \in \mathscr{H}_\theta, G_\Lambda \in \mathscr{H}_\Lambda^d} \frac{1}{N^2} \sum_{i,j} w_{ij}^\sigma (y_i - f_\theta(x_j) + \langle G_\Lambda(x_i), x_j - x_i \rangle)^2 \text{(Taylor-2)}, \qquad (18)$$

where $w_{ij}^\sigma = \exp\left(\frac{-\|x_i - x_j\|^2}{\sigma^2}\right)$, $\mathscr{H}_\theta$ is an RKHS ball of scalar valued functions, and $\mathscr{H}_\Lambda^d$ is an RKHS ball of vector valued functions.

Under mild assumptions, Mukherjee & Zhou (2006) shows that we have for $N = O(1/\epsilon^{d/2})$: $\int_\Omega \|G_\Lambda(x) - \nabla_x \psi(x)\|^2 \rho_\Omega(x) dx \leq \epsilon$. We simplify the problem in Eq. 18 and propose the following two objective functions and leverage the deep learning toolkit to parameterize the surrogate $f_\theta$:

$$\min_{f_\theta \in \mathscr{H}_\theta} \frac{1}{N^2} \sum_{i,j} w_{ij}^\sigma (y_i - f_\theta(x_j) + \langle \nabla_x f_\theta(x_i), x_j - x_i \rangle)^2 \text{(Taylor-1)}, \qquad (19)$$

$$\min_{f_\theta \in \mathscr{H}_\theta} \frac{1}{N} \sum_{i=1}^{N} \{ (y_i - f_\theta(x_i))^2 + \frac{\lambda}{N^2} \sum_{i,j} w_{ij}^\sigma (y_i - y_j + \langle \nabla_x f_\theta(x_i), x_j - x_i \rangle)^2 \}, \textbf{(Taylor-Reg)} . \quad (20)$$

The objective in Eq. 19 uses a single surrogate to parameterize the potential and its gradient. The objective in Eq. 20 is similar in spirit to the Jacobian matching formulation in the sense that it adds a regularizer on the gradient of the surrogate to be consistent with the first-order Taylor expansion in local neighborhoods. The advantage of the Taylor learning approach is that we do not need to perform zero-order estimation of gradients to construct the training set and we rely instead on first-order approximation in local neighborhood.

### 6.3 SURROGATE MODEL CONSTRAINED LMC

Consider the surrogate model $f_\theta$ obtained via Hermite Learning (Eq. 17) or via Taylor learning (Eqs 18, 19, 20). We are now ready to define the surrogate model LMC by replacing $\mathbf{G(x)} = \nabla_\mathbf{x} \mathbf{f}_\theta(\mathbf{x})$ in the constrained Langevin dynamics in Eqs 9 and 10.

Both Hermite and Taylor learning come with theoretical guarantees when the approximation function space is an RKHS under some mild assumptions on the training distribution and the regularity of the target function $\psi$. In Hermite learning (Theorem 2 in Shi et al. (2010)) we have: $E_{x \sim p_\Omega} \|\nabla_x f_\theta(x) - \nabla_x \psi(x)\|^2 \le \epsilon$ for sufficiently large training set $N = O(1/\epsilon^{1/(2\zeta r)})$ (where exponents $\zeta, r \in [0, 1]$ depend on regularity of $\psi$). In Taylor Learning with the objective function given in Eq. 18 (Proposition 7 in Wu et al. (2010) we have: $\mathbb{E}_{x \sim \rho_\Omega} \|G_\Lambda(x) - \nabla_x \psi(x)\|^2 \le \epsilon$ for $N = O(1/\epsilon^{d/2})$. In order to apply Theorem 2 we need this gradient approximation error to hold in expectation on all intermediate distributions in the Langevin sampling. Hence, we need the following extra-assumption on the training distribution $p_\Omega$:

**Assumption D**: Assume we have a learned surrogate $G$ on training distribution $\rho_\Omega$ such that $\mathbb{E}_{x \sim \rho_\Omega} \|G(x) - \nabla_x U(x)\|^2 \le \epsilon$. Assume $\rho_\Omega(x) > 0, \forall x \in \Omega$ and that it is a dominating measure of Langevin (PLMC, S-PLMC, Prox-LMC, S-ProxLMC ) intermediate distributions $\mu_k$, i.e. there exists $C > 0$ such that:

$$\mu_k(x) \le C\rho_\Omega(x), \forall x \in \Omega, \forall k = 0, \dots K - 1.$$

Under Assumption **D**, it follows immediately that

$$\mathbb{E} \|G(X_k) - \nabla U(X_k)\|^2 = \int_\Omega \|G(x) - \nabla U(x\|^2 \frac{\mu_k(x)}{\rho_\Omega(x)} \rho_\Omega(x) \le C\epsilon$$

and hence we can apply Theorem 2 for $\delta = C\epsilon$, and we obtain $\varepsilon$-approximation of the target Gibbs distribution in terms of total variation distance.

**Remark 4.** *Assumption D on the $\epsilon$-approximation of the gradient can be achieved for a large enough training set N, when we use Hermite learning in RKHS under mild assumptions and in Taylor learning. The assumption on the dominance of the training distribution is natural and means that we need a large training set that accounts to what we may encounter in Surrogate LMC iterations.*

In what follows we refer to surrogate constrained LMC, as **x-PLMC** or **x-ProxLMC** where x is one of four suffixes ({Z-Hermite, Taylor-2, Taylor-1, Taylor-Reg}).

## 7 RELATED WORK

**Zero-Order Methods.** Zero-order optimization with Gaussian smoothing was studied in Nesterov & Spokoiny (2017) and Duchi et al. (2015) in the convex setting. Non-convex zero order optimization was also addressed in Ghadimi & Lan (2013). The closest to our work is the zero-order Langevin Shen et al. (2019) introduced recently for black-box sampling from $\log$ concave density. The main difference in our setting, is that the density has a compact support and hence the need to appeal to projected LMC (Bubeck et al., 2015) and Proximal LMC (Brosse et al., 2017). It is worth nothing that Hsieh et al. (2018) introduced recently mirror Langevin sampling that can also be leveraged in our framework.

**Gradients and Score functions Estimators.** We used the approach of gradient distillation (Srinivas & Fleuret, 2018) and learning gradients of (Wu et al., 2010), since they are convenient for training on different constraints and they come with theoretical guarantees. However, other approaches can be also leveraged such as the score matching approach for learning the gradient of the log likelihood (Hyvärinen, 2005) and other variants appealing to dual embeddings (Dai et al., 2018). Estimating gradients can be also performed using Stein's method as in (Li & Turner, 2017), or via maintaining a surrogate of the gradient as in Stein descent without gradient (Han & Liu, 2018).

**Optimization approaches.** Due to space limitation, we restrict the discussion to the optimization methods that are most commonly and recently used for optimal material (or molecule) design. A popular approach to deal with optimization of expensive black-box functions is Bayesian Optimization (BO) (Mockus, 1994; Jones et al., 1998; Frazier, 2018). The standard BO protocol is comprised of estimating the black-box function from data through a probabilistic surrogate model, usually a Gaussian process, and maximizing an acquisition function to decide where to sample next. BO is often performed over a latent space, as in (Gómez-Bombarelli et al., 2018). Hernández-Lobato et al. (2016) proposed an information-theoretic framework for extending BO to address optimization under black-box constraints, which is close to current problem scenario. Genetic Algorithms (GA), a class of meta-heuristic based evolutionary optimization techniques, is another widely used approach for generating (material) samples with desired property (Jennings et al., 2019) and has been also used for handling optimization under constraints (Chehouri et al., 2016). However, GA typically requires a large number of function evaluations, can get stuck in local optima, and does not scale well with complexity. Finally, Zhou et al. (2019) has used deep reinforcement learning technique of Deep Q-networks to optimize molecules under a specific constraint using desired properties as rewards. The advantage of our framework is that we obtain a distribution of optimal configurations (as opposed to a single optimized sample) that does not rely on training on a specific pre-existing dataset and can be further screened and tested for their optimality for the task at hand.

## 8 EXPERIMENTS

In this section, we demonstrate the usability of our black-blox Langevin sampling approach for the design of nano-porous configurations. We first show the performance of the surrogate models in learning the potential function, showcasing the results using four different variants: standard regression, Taylor regularization, Taylor-1 and Taylor-2. We then show how well the surrogate-based Langevin MC generates new samples under the thermal and mechanical constraints. We compare the sample quality on multiple criteria between the surrogate and zero-order approaches with either projection or proximal update step.

**Data.** We want to learn surrogate models to approximate the gradient of the potential from data. To this end, we generate a dataset of 50K nano-porous structures, each of size 100nm $\times$ 100nm. One such example is displayed in Fig. 1. Number of pores is fixed to 10 in this study and each pore is a square with a side length of 17.32nm. We sample the pore centers uniformly over the unit square and construct the corresponding structure after re-scaling them appropriately. Then, using the solvers OpenBTE (Romano & Grossman, 2015) and Summit ($\sum$MIT Development Group, 2018), we obtain for each structure $x$ a pair of values: thermal conductivity $\kappa$ and von Mises stress $\sigma$. Finally, we collect two datasets: $\{(x_i, \kappa_i)\}_{i=1}^N$ and $\{(x_i, \sigma_i)\}_{i=1}^N$ with the same inputs $x_i$'s and $N = 50K$ samples. More details are given in Appendices B and C on the PDEs and their corresponding solvers.

**Features.** The pore locations are the natural input features to the surrogate models. Apart from the coordinates, we also derive some other features based on physical intuitions. For example, the distances between pores and the alignment along axes are informative of thermal conductivity (Romano & Grossman, 2016). As such, we compute pore-pore distances along each coordinate axis and add them as additional features.

**Surrogate gradient methods.** We use feed-forward neural networks to model the surrogates since obtaining gradients for such networks is efficient thanks to automatic differentiation frameworks. We use networks comprised of 4 hidden layers with sizes $128, 72, 64, 32$ and apply the same architecture to approximate the gradients for $\kappa$ and $\sigma$ separately. The hidden layers use ReLU activations whereas sigmoid was used at the output layer (after the target output is properly normalized). For the Taylor-2 variant (in Eq. 18), we have an additional output vector of the same size as the input for the gradient prediction. The networks are trained on the corresponding objective functions set up earlier by an

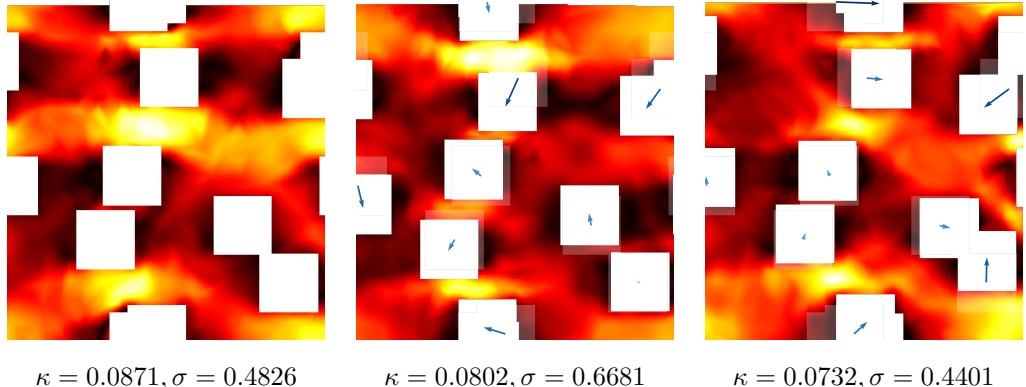

$$\kappa = 0.0871, \sigma = 0.4826 \qquad \kappa = 0.0802, \sigma = 0.6681 \qquad \kappa = 0.0732, \sigma = 0.4401$$

Figure 1: Example of nano-porous structures with corresponding heat flux shown using a color gradient. Yellow regions indicate high phonons flux. The thermal conductivity $\kappa$ and von Mises stress $\sigma$ are reported below each structure. The arrows show the moving directions of the pores. (Left) A random sample. (Middle) The sample obtained by Taylor-Reg PMLC starting from the left structure with $\kappa$ constraint. (Right) The sample obtained by Taylor-Reg PMLC with both $\kappa$ and $\sigma$ constraints.

Adam optimizer with learning rate $10^{-4}$ and decay 1.0. We fine-tune the networks with simple grid-search and select the best models for comparison. Due to the space constraint, we present the results in Appendix A and emphasize that Z-Hermite is not included in the entire comparison but in a small experiment performed with a more lightweight OpenBTE version.

**Incorporating constraints and comparison metrics.** We demonstrate the usability of our proposed black-box Langevin sampling for the design of nano-configurations under thermal conductivity and mechanical stability constraints that are provided by the corresponding PDE solvers. To compare sampling outcomes, we use the following metrics. We report the minimum value of $\kappa$ and Monte Carlo estimates for both $\kappa$ and $\sigma$ to compare the samples generated by different sampling methods and surrogate models. The Monte Carlo estimates are computed on 20 samples.

**Single constraint.** Our first task is to design nano-configurations under the thermal conductivity constraint where we want $\kappa$ as low as possible in order to achieve high thermo-electric efficiency. From the posterior regularization formulation Section 2, we pose the constraint satisfaction as sampling from the following Gibbs distribution:

$$\pi(x) = p_0(x) \frac{\exp(-\lambda \kappa(x)^2)}{Z} \mathbb{1}_{x \in [0,1]^{20}} \tag{21}$$

where $p_0(x)$ is the uniform distribution over the unit square, which is equivalent to the Poisson process of 10 pores on the square, and $\kappa(x)$ is the thermal conductivity we want to minimize. Starting from 20 samples initialized from $p_0(x)$, we run our proposed black-box Langevin MCs and obtain 20 new realizations from the target distribution $\pi(x)$. We use four different surrogates (including simple regression, Taylor-Reg, Taylor-1 and zero-order) and each surrogate with either projection or proximal update. We show the summary statistics of these samples in Table 1. The regression-PMLC in the first row and regression-ProxLMC in the fifth represent the sampling where the surrogate model are fitted on solely the mean square error objective. In all methods, we set $\lambda = 100$, the step size $\eta = 1e-3$ and the exponential decay rate 0.8. Since keeping track of the true $\kappa$ value is expensive, we stop after $K = 10$ iterations. We first observe that the regression-based method (PMLC, ProxLMC) is less effective than the others simply because they do not have an implicit objective for approximating the gradients. Taylor-Reg and Taylor-1 demonstrate its effectiveness in approximating the gradient and are able to achieve lower thermal conductivity. In particular, Taylor-1-ProxLMC and Zero-order-PMLC perform in the similar range in terms of the minimum achieved, but the learned surrogate offers **17x** speed up (per sample) over zero order methods. Due to the space limit, we do not report Taylor-2 results in Table 1, and note that Taylor-2 works in the similar vein as Taylor-1.

| Model | Min $\kappa$ | Mean $\kappa$ | Per-sam. time (s) |
|---|---|---|---|
| Regression-PLMC | 0.0757 | $0.1206 \pm 0.0480$ | 1055 |
| Taylor-Reg-PLMC | 0.0638 | $0.1196 \pm 0.0495$ | 899 |
| Taylor-1-PLMC | 0.0637 | $0.1278 \pm 0.0610$ | 852 |
| Zero-order-PLMC | **0.0510** | **$0.1093 \pm 0.0271$** | **14967** |
| Regression-ProxLMC | 0.0646 | $0.1282 \pm 0.0531$ | 1107 |
| Taylor-Reg-ProxLMC | 0.0712 | $0.1205 \pm 0.0455$ | 899 |
| Taylor-1-ProxLMC | **0.0575** | **$0.1297 \pm 0.0543$** | **874** |
| Zero-order-ProxLMC | 0.0719 | $0.1112 \pm 0.0363$ | 14938 |

Table 1: Statistics of 20 new samples obtained by running different surrogate-based Langevin MCs on $\pi$ with the thermal conductivity constraint (Eq. 21). We show the min and mean over the generated samples and the per-sample time. Initialized samples have min $\kappa = 0.0619$ and mean $\sigma = 0.1268$.

**Multiple constraints.** Achieving the minimal thermal conductivity can be fulfilled without much difficulty (e.g. structures with all pores aligned along the vertical axis), but such structures are often mechanically unstable. In the next step, we study whether adding more (conflicting) constraints helps us design better nano-configurations. Hence, we consider both thermal conductivity $\kappa$ and mechanical stability provided via von Mises stress $\sigma$. We want a sample $x$ that minimizes $\kappa(x)$ to achieve high thermo-electric efficiency while maintaining $\sigma(x)$ less than some threshold (which we explain below). Like the single constraint case, we pose this as sampling from the following Gibbs distribution:

$$\pi(x) = p_0(x) \frac{\exp(-\lambda_1 \kappa(x)^2 - \lambda_2 [\sigma(x) - \tau]_+)}{Z} \mathbb{1}_{x \in [0,1]^{20}}, \qquad (22)$$

where $p_0(x)$ is the same as above, $\sigma(x)$ is the von Mises stress and $\tau$ is a threshold on the maximum value of $\sigma$. With this framework, we relax the inequality constraint to the Hinge loss term on von Mises stress. The results are summarized in Table 2. Note that all the surrogate Langevin MCs are initialized from the same set of 20 samples as above. In this experiment, we set $\tau = 0.5$, $\lambda_1 = 100$, $\lambda_2 = 10$ the step size $\eta = 1e-3$ and the exponential decay rate $0.8$. Comparing with Table 1, one can see that not only better $\kappa$ be achieved but also the $\sigma$ can be reduced simultaneously. These results suggest that our approach can effectively sample new configurations under multiple competing constraints. Examples of new nano-configurations are show in Fig. 1 and Appendix A Fig. 5, 6 and 7.

| Model | Min $\kappa$ | Mean $\kappa$ | Mean $\sigma$ | Per-sam. time (s) |
|---|---|---|---|---|
| Taylor-Reg-PLMC | 0.0613 | $0.1256 \pm 0.0538$ | $0.6590 \pm 0.2261$ | 952 |
| Taylor-1-PLMC | 0.0611 | $0.1278 \pm 0.0610$ | $0.6380 \pm 0.1598$ | 852 |
| Zero-order-PLMC | 0.0471 | $0.1148 \pm 0.0475$ | $0.6511 \pm 0.1916$ | **15677** |
| Taylor-Reg-ProxLMC | 0.0666 | $0.1195 \pm 0.0534$ | $0.6402 \pm 0.1464$ | 856 |
| Taylor-1-ProxLMC | **0.0548** | $0.1298 \pm 0.0610$ | $0.6156 \pm 0.1463$ | **972** |
| Zero-order-ProxLMC | **0.0354** | **$0.1080 \pm 0.0384$** | **$0.6029 \pm 0.1376$** | **15080** |

Table 2: Summary statistics of 20 new samples obtained by our sampling method on $\pi(x)$ with $\kappa$ and $\sigma$ constraints Eq. 22. The starting samples are reused from the single constraint case (min $\kappa = 0.0759$, mean $\kappa = 0.1268$, and mean $\sigma = 0.8181$; note that $\sigma$ can be as high as 16.)

## 9 CONCLUSION

In this paper we introduced Surrogate-Based Constrained Langevin Sampling for black-box sampling from a Gibbs distribution defined on a compact support. We studied two approaches for defining the surrogate: the first through zero-order methods and the second via learning gradient approximations using deep neural networks. We showed the proofs of convergence of the two approaches in the log-concave and smooth case. While zero-order Langevin had prohibitive computational cost, learned surrogate model Langevin enjoy a good tradeoff of lightweight computation and approximation power. We applied our black-box sampling scheme to the problem of nano-material configuration design, where the black box constraints are given by expensive PDE solvers, and showed the efficiency and the promise of our method in finding optimal configurations. Among different approaches for approximating the gradient, the zero-order ones (PLMC, ProxLMC) show overall superior performance, at a prohibitive computational cost. We established that the deep the surrogate (Taylor-1

ProxLMC) is a viable alternative to zero-order methods, achieving reasonable performance, and offering **15x** speedup over zero-order methods.

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

## A    SUPPLEMENTAL EXPERIMENTAL RESULTS

**Surrogate gradient methods** We use feed-forward neural networks to model the surrogates since obtaining gradients for such networks is efficient thanks to automatic differentiation frameworks. We use networks comprised of 4 hidden layers with sizes $128, 72, 64, 32$ and apply the same architecture to approximate the gradients for $\kappa$ and $\sigma$ separately. The hidden layers compute ReLU activation whereas sigmoid was used at the output layer (after the target output is properly normalized). For the Taylor-2 variant (in Eq. 18), we have an output vector for the gradient prediction. The networks are trained on the corresponding objective functions set up earlier by Adam optimizer with learning rate $10^{-4}$ and decay $1.0$. We fine-tune the networks with simple grid-search and select the best models for comparison.

As emphasized throughout, our focus is more on approximating the gradient rather than learning the true function. However, we need to somehow evaluate the surrogate models on how well they generalize on a hold-out test set. Like canonical regression problems, we compare the surrogate variants against each other using root mean square error (RMSE) on the test set. Figures 2 and 3 shows the results. The left figure shows RMSE for predicting $\kappa$ and the right one shows RMSE for the von Mises stress $\sigma$. We can see that the Taylor-Reg generalizes better and also converges faster than Taylor-1 and Taylor-2 to target RMSE for $\kappa$, while all methods result similarly for $\sigma$ prediction. This is reasonable because the objectives of Taylor-1 and Taylor-2 are not to optimize the mean square error, which we evaluate on here. Figure 3 shows the learning in terms of sample complexity. Again, Taylor-Reg outperforms Taylor-1 and Taylor-2 for $\kappa$ prediction. In contrast, most models work similarly for $\sigma$ regression, particularly when the training size is reduced to 50% (25K).

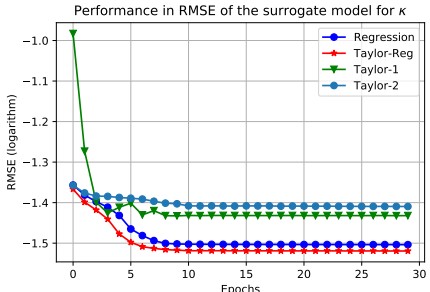 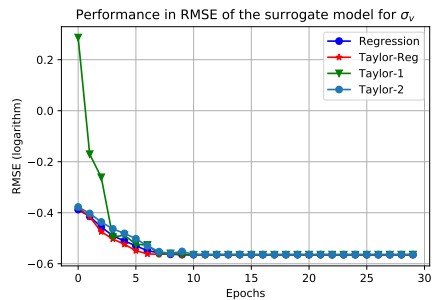

Figure 2: Comparison of the surrogate variants in testing RMSE. (Left) prediction accuracy for the thermal conductivity $\kappa$. (Right) prediction accuracy for mechanical stability $\sigma$. Note the difference in scale of $\kappa$ and $\sigma$.

**Effectiveness of Z-Hermite learning** Notice that Z-Hermite learning is not included in this comparison and as a surrogate model in the black-blox Langevin sammpling in Section 8. The reason is that apart from the usual sample pair $(x_i, y_i)$, we need the gradient $\tilde{y}_i$ (See Eq. 17). Since we can query the solvers, this gradient can only be estimated using finite difference. For both $\kappa$ and $\sigma$ in our experiment, obtaining such data is extremely expensive. As a consequence, we do not have the full results of the Z-Hermite model. Instead, we ran a separate study to show the effectiveness of Z-Hermite surrogate LMC on a smaller data with a lightweight OpenBTE version (0.9.55). The results in Table 3 shows the working of Z-Hermite learning in learning the gradient of $\kappa(x)$. Here, the entropy is based nearest neighbor estimate to demonstrate the diversity of the pore centers in the unit square. With the $(x_p, y_p)$-coordinates of each pore $p$, the entropy estimate is given by:

$$H = \frac{1}{n} \sum_{i=1}^{n} \log(n \min_{j \neq i} \|p_i - p_j\|) + \log 2 + C.$$

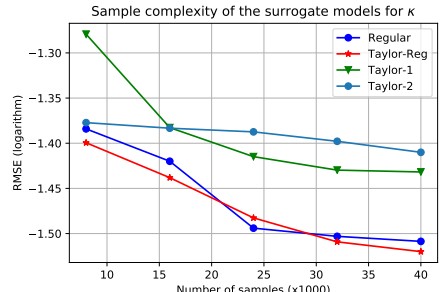 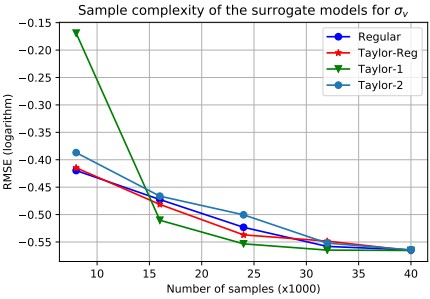

Figure 3: Comparison of the surrogate models in RMSE on the same test set when the training size is varied. Note the scale difference in the figures due to the different range of values.

| Model | Mean $\kappa$ | Mean entropy | Per-sam. time (s) |
|---|---|---|---|
| Zero-order PLMC | **0.0676** | **1.960** | 3658 |
| Taylor-1 PLMC | 0.0988 | 1.745 | 253 |
| Z-Hermite PLMC | 0.0946 | 1.739 | 227 |
| Hybrid (Zero-order + Taylor-1) | 0.0786 | 1.867 | 2136 |

Table 3: Z-Hermite learning is sample efficient at training time as well as for the Langevin sampling, but collecting the training set is prohibitive. Zero-order and deep surrogate can work in a hybird way and offer a better trade-off between accuracy and computation.

**A hybrid algorithm between zero-order and Taylor-1 surrogate** We can see in Tables 1, 2 and 3 the trade-off between computation and accuracy of our approach. While zero-order PLMC and ProxLMC can achieve the lowest thermal conductivity, their computational costs are prohibitive. In contrast, deep surrogate models (including Taylor-Reg, Taylor-1) are far more time-efficient but slightly worse in terms of achieving the optimal $\kappa$. To mitigate the trade-off, we propose a simple hybrid method that combines the best of the zero-order and Taylor-1 surrogate models. The algorithm is shown in Figure A that alternates between using the gradient from the zero-order estimate and the gradient of the deep surrogate depending on whether taking this step would decrease the potential function (i.e. $\kappa$). We show and compare the achieved $\kappa$ and running time in Table 3. Examples of the samples generated by Zero-order PLMC, Taylor-1 PLMC and the hybrid method are also depicted in Figure 4. The hybrid achieves the thermal conductivity that is lower than Taylor-1 PMLC while running almost **2x** faster than zero-order PLMC. This suggests that the hybrid strategy offers a better trade-off in accuracy and computation. One way to further improve the hybrid is to collect the zero-order gradients while mixing and re-update the surrogate with Z-Hermite learning.

**Algorithm 1** A hybrid PLMC algorithm alternating between zero-order and Taylor-1 surrogate gradients.

Train a network $f_\theta(x)$ with Taylor-1
Randomly sample $x_0$ from the uniform $p(x)$
Perform a Langevin dynamic step
**for** $t = 1, 2, \ldots, K$ **do**
    **if** $\kappa(x - \eta\nabla_x f_\theta(x)) < \kappa(x)$ **then**
        $x \leftarrow P_\Omega(x - \eta\nabla_x f_\theta(x) + \sqrt{2\eta}\xi)$
    **else**
        estimate $\widetilde{\nabla}\kappa(x)$ using zero-order method
        update $x \leftarrow P_\Omega(x - \eta\widetilde{\nabla}\kappa(x) + \sqrt{2\eta}\xi)$
    **end if**
**end for**
Return a new sample $x$

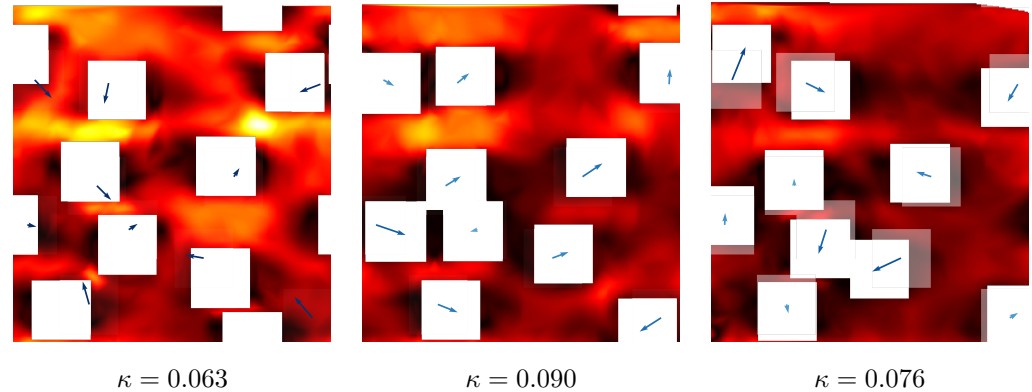

$$\kappa = 0.063 \qquad\qquad \kappa = 0.090 \qquad\qquad \kappa = 0.076$$

Figure 4: Samples from by Zero-order PLMC (left), Taylor-1 PLMC (middle) and the hybrid algorithm of Zero-order and Taylor-1 PLMC (right). All are run with the $\kappa$ constraint.

**Additional generated samples** We show additional configurations generated by our sampling approach (Taylor-Reg ProxLMC, Taylor-1 ProxLMC and Zero-order ProxLMC) in Fig. 5, 6 and 7.

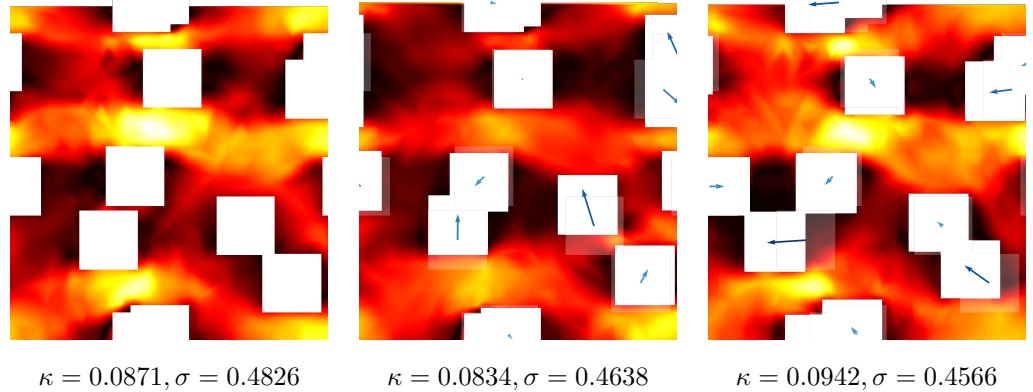

$$\kappa = 0.0871, \sigma = 0.4826 \qquad \kappa = 0.0834, \sigma = 0.4638 \qquad \kappa = 0.0942, \sigma = 0.4566$$

Figure 5: Example of nano-porous structures with corresponding heat flux shown using a color gradient. Yellow regions indicate high phonons flux. The thermal conductivity $\kappa$ and von Mises stress $\sigma$ are reported below each structure. The arrows show the moving directions of the pores. (Left) A random sample. (Middle) The sample obtained by Taylor-Reg ProxLMC starting from the left structure with $\kappa$ constraint. (Right) The sample obtained by Taylor-Reg ProxLMC with both $\kappa$ and $\sigma$ constraints.

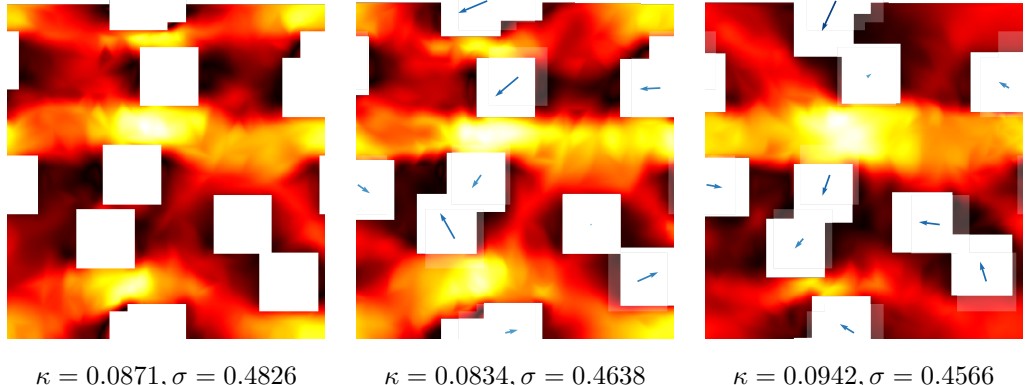

$$\kappa = 0.0871, \sigma = 0.4826 \qquad \kappa = 0.0834, \sigma = 0.4638 \qquad \kappa = 0.0942, \sigma = 0.4566$$

Figure 6: Example of nano-porous structures with corresponding heat flux shown using a color gradient. Yellow regions indicate high phonons flux. The thermal conductivity $\kappa$ and von Mises stress $\sigma$ are reported below each structure. The arrows show the moving directions of the pores. (Left) A random sample. (Middle) The sample obtained by Taylor-1 ProxLMC starting from the left structure with $\kappa$ constraint. (Right) The sample obtained by Taylor-1 ProxLMC with both $\kappa$ and $\sigma$ constraints.

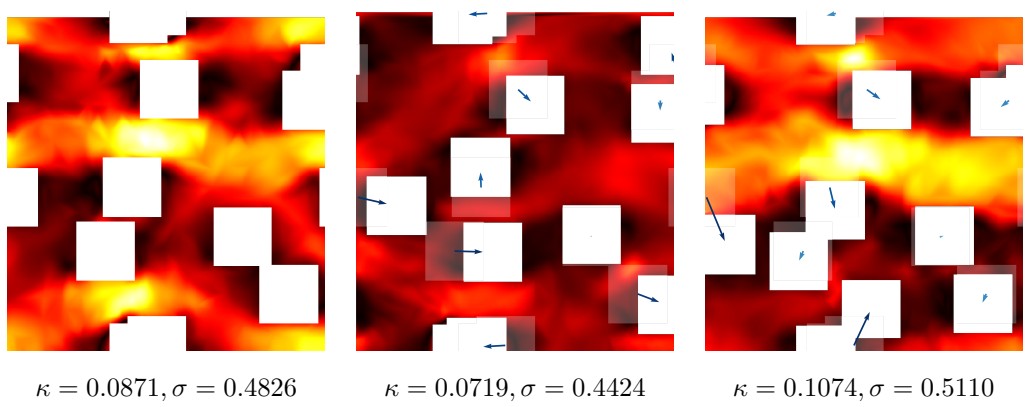

$$\kappa = 0.0871, \sigma = 0.4826 \qquad \kappa = 0.0719, \sigma = 0.4424 \qquad \kappa = 0.1074, \sigma = 0.5110$$

Figure 7: Example of nano-porous structures with corresponding heat flux shown using a color gradient. Yellow regions indicate high phonons flux. The thermal conductivity $\kappa$ and von Mises stress $\sigma$ are reported below each structure. The arrows show the moving directions of the pores. (Left) A random sample. (Middle) The sample obtained by Zero-order ProxLMC starting from the left structure with $\kappa$ constraint. (Right) The sample obtained by Zero-order ProxLMC with both $\kappa$ and $\sigma$ constraints.

## B  BACKGROUND ON MODELING NANOSCALE HEAT TRANSPORT

At the nanoscale, heat transport may exhibit strong ballistic behaviour and a non-diffusive model must be used (Chen, 2005). In this work we use the Boltzmann transport equation under the relaxation time approximation and in the mean-free-path (MFP) formulation (Romano & Grossman, 2015)

$$\Lambda \hat{\mathbf{s}} \cdot \nabla T(\Lambda) + T(\Lambda) = \int \alpha(\Lambda') \langle T(\Lambda') \rangle d\Lambda', \tag{23}$$

where $T(\Lambda)$ is the effective temperature associated to phonons with MFP $\Lambda$ and direction $\hat{\mathbf{s}}$; the notation $\langle . \rangle$ stands for an angular average. The coefficients $\alpha(\Lambda')$ are given by

$$\alpha(\Lambda') = \frac{K(\Lambda')}{\Lambda'} \left[ \int \frac{K(\Lambda'')}{\Lambda''} d\Lambda'' \right]^{-1}, \tag{24}$$

where $K(\Lambda')$ is the bulk MFP distribution. In general, such a quantity can span several orders of magnitude; however, for simplicity we assume the *gray* model, i.e. all phonons travel with the same MFP, $\Lambda_0$. Within this approximation, we have $K(\Lambda) = \kappa_{\text{bulk}} \delta(\Lambda - \Lambda_0)$. In this work we choose $\Lambda_0$ = 10 nm, namely as large as the unit cell, so that significant phonons size effects occur. With no loss of generality, we set $\kappa_{\text{bulk}} = 1 \ \text{Wm}^{-1}\text{K}^{-1}$ . Eq. 23 is an integro-differential PDE, which is solved iteratively for each phonon direction over an unstructured mesh (Romano & Di Carlo, 2011). We apply periodic boundary conditions along the unit cell while imposing a difference of temperature of $\Delta T$ = 1 K along the *x*-axis. At the pores' walls we apply diffusive boundary conditions. Upon convergence, the effective thermal conductivity is computed using Fourier's law, i.e.

$$\kappa_{\text{eff}} = -\frac{L}{\Delta T A} \int_A \mathbf{J} \cdot \hat{\mathbf{n}} dS, \tag{25}$$

where $\mathbf{J} = (\kappa_{\text{bulk}}/\Lambda_0)\langle T(\Lambda_0)\hat{\mathbf{s}}\rangle\hat{\mathbf{n}}$ is the heat flux, $L$ is the size of the unit cell, $A$ is the area of the cold contact (with normal $\hat{\mathbf{n}}$). Throughout the text we use the quantity $\kappa = \kappa_{\text{eff}}/\kappa_{\text{bulk}}$ as a measure of phonon size effects.

## C   BACKGROUND ON MODELING MECHANICAL STRESS

We model mechanical stress by using the continuum linear elasticity equations

$$\frac{\partial}{\partial x_j} \sigma_{ij} = f_i, \tag{26}$$

where $f_i$ is the body force (which is zero in this case), and $\sigma_{ij}$ is the stress tensor. Note that we used the Einstein notation, i.e. repeated indexes are summed over. The strain $\epsilon_{kl}$ is related to the stress via the fourth-rank tensor elastic constant $C_{ijkl}$

$$\sigma_{ij} = C_{ijkl}\epsilon_{kl}. \tag{27}$$

The strain is then related to the displacement $\mathbf{u}$ via

$$\epsilon_{kl} = \frac{1}{2}\left(\frac{\partial u_k}{\partial x_l} + \frac{\partial u_l}{\partial u_k}\right). \tag{28}$$

We apply periodic boundary conditions along the unit-cell and applied solicitation is a small in-plane expansion. Once the stress tensor is calculated, we compute the von Mises stress as

$$\sigma_{VM} = \sqrt{\frac{1}{2}(\sigma_3 - \sigma_2)^2 + (\sigma_3 - \sigma_1)^2 + (\sigma_2 - \sigma_1)^2}, \tag{29}$$

where $\sigma_i$ are the principal stress axis. As a mechanical stability estimator we use $\sigma = \max_{\mathbf{x} \in D}(\sigma_{VM})$ where $D$ is the simulation domain. To avoid material's plasticity, $\sigma$ needs to be smaller than the yield stress of a given material. For mechanical simulation we used the SUMIT code ($\sum$MIT Development Group, 2018).

## D   BACKGROUND ON STOCHASTIC DIFFERENTIAL EQUATIONS (SDE): CHANGE OF MEASURE AND GRISANOV'S FORMULA

**Theorem 3** (Grisanov Theorem, Change of Measure for Brownian Motion (Lipster & Shiryaev, 2001), Theorem 6.3 page 257)**.** *Let $(W_t, \mathscr{F}_t)$ be a Wiener process (Brownian motion) and $(\beta_t, \mathscr{F}_t)$ a random process such that for any $T > 0$*

$$\int_0^T \|\beta_t\|^2 dt < \infty \ a.s$$

*Then the random process : $d\tilde{W}_t = dW_t - \beta_t dt$ or written equivalently: $\tilde{W}_t = W_t - \int_0^t \beta_s ds$, is a Wiener process with respect to $\mathscr{F}_t$, $t \in [0, T]$. Let $P_T^W = \mathscr{L}(W_{[0,T]})$, and $P_T^{\tilde{W}} = \mathscr{L}(\tilde{W}_{[0,T]})$ the densities are given by: $\frac{dP_T^{\tilde{W}}}{dP_T^W} = \exp\left(\int_0^T \langle \beta_s, dW_s \rangle - \frac{1}{2}\int_0^T \|\beta_s\|^2 ds\right)$. It follows that:*

$$KL(P_T^W, P_T^{\tilde{W}}) = \frac{1}{2}\mathbb{E}_{P_T^W}\left[\int_0^T \|\beta_s\|^2 ds\right] \tag{30}$$

**Theorem 4** (Grisanov Theorem, Change of Measure for Diffusion Processes, (Lipster & Shiryaev, 2001), ()). *Let $(X_t)_{t \geq 0}$ and $(Y_t)_{t \geq 0}$*

$$dX_t = \alpha_t(X)dt + dW_t$$

$$dY_t = \beta_t(Y)dt + dW_t$$

*where $X_0 = Y_0$ is an $\mathscr{F}_0$ measurable random variable. Suppose that the non-anticipative functionals $\alpha_t(x)$ and $\beta_t(x)$ are such that a unique continuous strong solutions exits for both processes. If for any $T > 0$:*

$$\int_0^T \|\alpha_s(X)\|^2 + \|\beta_s(X)\|^2 ds < \infty (a.s) \text{ and } \int_0^T \|\alpha_s(Y)\|^2 + \|\beta_s(Y)\|^2 ds < \infty (a.s).$$

*Let $P_T^X = \mathscr{L}(X_{[0,T]})$, and $P_T^Y = \mathscr{L}(Y_{[0,T]})$.*

$$\frac{dP_T^Y}{dP_T^X}(X) = \exp\left(-\int_0^T \langle \alpha_s(X) - \beta_s(X), dX_s \rangle + \frac{1}{2}\int_0^T (\|\alpha_s(X)\| - \|\beta_s(X)\|^2)ds\right).$$

$$KL(P_T^X, P_T^Y) = \frac{1}{2}\mathbb{E}_{P_T^X}\left[\int_0^T \|\alpha_s(X) - \beta_s(X)\|^2 ds\right]. \tag{31}$$

## E BACKGROUND ON ZERO-ORDER OPTIMIZATION (GRADIENT-FREE)

Consider the smoothed potential $U_\nu$ defined as follows:

$$U_\nu(x) = \mathbb{E}_{g \sim \mathscr{N}(0, I_d)} U(x + \nu g)$$

its gradient is given by:

$$\nabla_x U_\nu(x) = \mathbb{E}_g \frac{U(x + \nu g) - U(x)}{\nu} g,$$

A monte carlo estimate of $\nabla_x U_\nu(x)$ is:

$$\hat{G}_n(x) = \frac{1}{n}\sum_{j=1}^n \left(\frac{U(x + \nu g_j) - U(x)}{\nu}\right) g_j,$$

where $g_1, \ldots g_n$ are iid standard Gaussians vectors.

Using known results in zero order optimization under assumptions on smoothness and bounded gradients of the gradients we have for all $x$ ((Nesterov & Spokoiny, 2017; Shen et al., 2019)):

$$\mathbb{E}_g \left\|\hat{G}_1(x) - \nabla_x U(x)\right\|^2 \leq \left(\beta\nu(d+2)^{3/2} + (d+1)^{\frac{1}{2}}\|\nabla_x U(x)\|\right)^2 \leq \left(\beta\nu(d+2)^{3/2} + (d+1)^{\frac{1}{2}}L\right)^2$$

Finally by independence of $u_1, \ldots u_n$ we have:

$$\mathbb{E}_{g_1,\ldots,g_n} \left\|\hat{G}_n(x) - \nabla_x U(x)\right\|^2 \leq \frac{\left(\beta\nu(d+2)^{3/2} + (d+1)^{\frac{1}{2}}L\right)^2}{n} \tag{32}$$

## F  PROOFS

*Proof of Lemma 1.* Define the Lagrangian:

$$L(q, \eta) = \int_\Omega \log\left(\frac{q(x)}{p_0(x)}\right) q(x) dx + \sum_{j=1}^{C_e} \lambda_j \int_\Omega (\psi_j(x) - y_j)^2 q(x) dx$$

$$+ \sum_{k=1}^{C_i} \lambda_k \int_{x \in \Omega} (\phi_k(x) - b_k)_+ q(x) dx + \eta \left(1 - \int_{x \in \Omega} q(x)\right)$$

Setting first order optimality conditions on $q$, we have for $x \in \Omega$:

$$\log\left(\frac{q(x)}{p_0(x)}\right) + 1 + \sum_{j=1}^{C} \lambda_j (\psi_j(x) - y_j)^2 + \sum_{k=1}^{C_i} \lambda_k (\phi_k(x) - b_k)_+ - \eta = 0$$

Hence we have:

$$q(x) = p_0(x) \frac{\exp\left(-\sum_{j=1}^{C_e} \lambda_j (\psi_j(x) - y_j)^2 - \sum_{k=1}^{C_i} \lambda_k (\phi_k(x) - b_k)_+\right)}{e \exp -\eta}, x \in \Omega$$

and

$$q(x) = 0, x \notin \Omega,$$

First order optimality on $\eta$ give us: $\int_\Omega q(x) = 1$, we conclude by setting $e \exp(-\eta) = Z$. $\square$

*Proof of Theorem 1 1) Projected Langevin.* Let us define the following continuous processes by interpolation of $X_k$ and $Y_K$ (Piecewise constant):

$$d\tilde{X}_t = P_\Omega(\tilde{U}_t(\tilde{X})dt + \sqrt{2\lambda}dW_t)$$

where $\tilde{U}_t(\tilde{X}) = -\sum_{k=0}^{\infty} \nabla_x U(\tilde{X}_{k\eta}) \mathbb{1}_{t \in [k\eta, (k+1)\eta]}(t)$. Similarly let us define :

$$d\tilde{Y}_t = P_\Omega(G_t(\tilde{Y})dt + \sqrt{2\lambda}dW_t)$$

where $G_t(\tilde{Y}) = -\sum_{k=0}^{\infty} G(\tilde{Y}_{k\eta}) \mathbb{1}_{t \in [k\eta, (k+1)\eta]}(t)$.

It is easy to see that we have : $X_k = \tilde{X}_{k\eta}$ and $Y_k = \tilde{Y}_{k\eta}$.

Let $\pi_{\tilde{X}}^T$ and $\pi_{\tilde{Y}}^T$ be the distributions of $(\tilde{X}_t)_{t \in [0,T]}$ and $(\tilde{Y})_{t \in [0,T]}$.

Note that :

$$d\tilde{Y}_t = P_\Omega \left(\tilde{U}_t(\tilde{X}_t)dt + \sqrt{2\lambda}(dW_t + \frac{1}{\sqrt{2\lambda}}(G_t(\tilde{Y}_t) - \tilde{U}_t(\tilde{X}_t))dt)\right)$$

Let

$$d\tilde{W}_t = dW_t + \frac{1}{\sqrt{2\lambda}}(G_t(\tilde{Y}_t) - \tilde{U}_t(\tilde{X}_t))dt$$

Hence we have :

$$d\tilde{Y}_t = P_\Omega \left(\tilde{U}_t(\tilde{X}) + \sqrt{2\lambda}d\tilde{W}_t\right),$$

Assume that $X_0 = Y_0$ there exists $\mathcal{Q}$ such that , $X_T = \mathcal{Q}(\{W_t\}_{t \in [0,T]})$ and $Y_T = \mathcal{Q}((\tilde{W}_t)_{t \in [0,T]})$. Let $\mu_T^{\tilde{X}}$ be the law of $\tilde{X}_{t \in [0,T]}$. Same for $\mu_T^{\tilde{Y}}$. The proof here is similar to the proof of Lemma 8 in (Bubeck et al., 2015). By the data processing inequality we have:

$$\text{KL}(\mu_T^{\tilde{X}}, \mu_T^{\tilde{Y}}) \leq \text{KL}(W_{t \in [0,T]}, \tilde{W}_{t \in [0,T]}),$$

Now using Grisanov's Theorem for change of measure of Brownian Motion (Theorem 3) we have:

$$\text{KL}(W_{t \in [0,T]}, \tilde{W}_{t \in [0,T]}) = \frac{1}{4\lambda} \mathbb{E} \int_0^T |G_t(\tilde{Y}_t) - \tilde{U}_t(\tilde{X}_t)|^2 dt$$

Consider $T = K\eta$, hence we have (with some abuse of notation we drop tilde as $Y_k = \tilde{Y}_{k\eta}$):

$$
\begin{aligned}
\mathrm{KL}(\mu_T^{\tilde{X}}, \mu_T^{\tilde{Y}}) &\leq \frac{1}{4\lambda} \mathbb{E} \int_0^{K\eta} |G_t(\tilde{Y}_t) - \tilde{U}_t(\tilde{X}_t)|^2 dt \\
&= \frac{1}{4\lambda} \mathbb{E} \sum_{k=0}^{K-1} \int_{k\eta}^{(k+1)\eta} \|G(Y_{k\eta}) - \nabla_x U(X_{k\eta})\|^2 dt \\
&= \frac{\eta}{4\lambda} \sum_{k=0}^{K-1} \mathbb{E} \|G(Y_{k\eta}) - \nabla_x U(X_{k\eta})\|^2 \\
&= \frac{\eta}{4\lambda} \sum_{k=0}^{K-1} \mathbb{E} \|G(Y_{k\eta}) - \nabla_x U(Y_{k\eta}) + \nabla_x U(Y_{k\eta}) - \nabla_x U(X_{k\eta})\|^2 \\
&\leq \frac{\eta}{2\lambda} \sum_{k=0}^{K-1} \left( \mathbb{E} \|G(Y_{k\eta}) - \nabla_x U(Y_{k\eta})\|^2 + \mathbb{E} \|\nabla_x U(Y_{k\eta}) - \nabla_x U(X_{k\eta})\|^2 \right)
\end{aligned}
$$

where in the last inequality we used the fact that $||a - b||^2 \leq 2(||a||^2 + ||b||^2)$. Note that we have by smoothness assumption on $U$:

$$
\|\nabla_x U(Y_{k\eta}) - \nabla_x U(X_{kh})\|^2 \leq \beta^2 \|X_{kh} - Y_{kh}\|^2
$$

Let $R$ be the diameter of $\Omega$, we can get a bound as follows:

$$
\begin{aligned}
\mathrm{KL}(\mu_T^{\tilde{X}}, \mu_T^{\tilde{Y}}) &\leq \frac{\eta}{2\lambda} \left( \underbrace{\sum_{k=0}^{K-1} \mathbb{E} \|G(Y_{k\eta}) - \nabla_x U(Y_{k\eta})\|^2}_{\text{Gradient approximation error}} + \beta^2 \sum_{k=0}^{K-1} \mathbb{E} \|X_{kh} - Y_{kh}\|^2 \right) \\
&\leq \frac{\eta}{2\lambda} \left( \sum_{k=0}^{K-1} \mathbb{E} \|G(Y_{k\eta}) - \nabla_x U(Y_{k\eta})\|^2 + K\beta^2 R^2 \right)
\end{aligned}
$$

Now using Pinsker inequality we have:

$$
TV(\mu_T^{\tilde{X}}, \mu_T^{\tilde{Y}})^2 \leq 2\mathrm{KL}(\mu_T^{\tilde{X}}, \mu_T^{\tilde{Y}}) \leq \frac{\eta}{\lambda} \left( \sum_{k=0}^{K-1} \mathbb{E} \|G(Y_{k\eta}) - \nabla_x U(Y_{k\eta})\|^2 + K\beta^2 R^2 \right)
$$

Hence for $T = K\eta$ we have:

$$
TV(\mu_K^{\text{S-PLMC}}, \mu_K^{\text{PLMC}}) \leq \sqrt{\frac{\eta}{\lambda} \left( \sum_{k=0}^{K-1} \mathbb{E} \|G(Y_k) - \nabla_x U(Y_k)\|^2 + K\beta^2 R^2 \right)^{\frac{1}{2}}}. \tag{33}
$$

$\square$

*Proof of Theorem 1 2) Proximal LMC.* Let us define the following continuous processes by interpolation of $X_k$ and $Y_K$ (Piecewise constant):

$$
d\tilde{X}_t = \tilde{U}_t(\tilde{X})dt + \sqrt{2\lambda}dW_t
$$

where $\tilde{U}_t(\tilde{X}) = -\sum_{k=0}^{\infty}(\nabla_x U(\tilde{X}_{k\eta}) + \frac{1}{\gamma}(\tilde{X}_{k\eta} - P_\Omega(\tilde{X}_{k\eta})))\mathbb{1}_{t \in [k\eta, (k+1)\eta]}(t)$. Similarly let us define :

$$
d\tilde{Y}_t = G_t(\tilde{Y})dt + \sqrt{2\lambda}dW_t
$$

where $G_t(\tilde{Y}) = -\sum_{k=0}^{\infty}(G(\tilde{Y}_{k\eta}) + \frac{1}{\gamma}(\tilde{Y}_{k\eta} - P_{\Omega}(\tilde{Y}_{k\eta})))\mathbb{1}_{t \in [k\eta, (k+1)\eta]}(t)$. Now applying Grisanov's Theorem for diffusions (Theorem 4) we have:

$$\text{KL}(\mu_T^{\tilde{X}}, \mu_T^{\tilde{Y}}) = \frac{1}{4\lambda}\mathbb{E}_{P_T^X}\left[\int_0^T \left\|U_t(\tilde{X}) - G_t(\tilde{X})\right\|^2 dt\right]$$

$$= \frac{1}{4\lambda}\mathbb{E}\sum_{k=0}^{K-1}\int_{k\eta}^{(k+1)\eta}\left\|G(\tilde{X}_{k\eta}) - \nabla_x U(\tilde{X}_{k\eta})\right\|^2 dt$$

$$= \frac{\eta}{4\lambda}\sum_{k=0}^{K-1}\mathbb{E}\left\|G(\tilde{X}_{k\eta}) - \nabla_x U(\tilde{X}_{k\eta})\right\|^2$$

$$= \frac{\eta}{4\lambda}\sum_{k=0}^{K-1}\mathbb{E}\left\|G(X_k) - \nabla_x U(X_k)\right\|^2.$$

Now using Pinsker inequality we have:

$$TV(\mu_{\tilde{X}}^T, \mu_{\tilde{Y}}^T)^2 \leq 2\text{KL}(\mu_{\tilde{X}}^T, \mu_{\tilde{Y}}^T).$$

Hence for $T = K\eta$ we have:

$$TV(\mu_K^{\text{S-ProxLMC}}, \mu_K^{\text{ProxLMC}}) \leq \sqrt{\frac{\eta}{2\lambda}}\left(\sum_{k=0}^{K-1}\mathbb{E}\left\|G(X_k) - \nabla_x U(X_k)\right\|^2\right)^{\frac{1}{2}}. \tag{34}$$

$\square$

*Proof of Theorem 2.* **S-PLMC.** If we set $\lambda = 1$, $\eta \leq \alpha/K^2$, where $\alpha = 1/(\delta + \beta^2 R^2)$, in this Corollary we obtain that : $TV(\mu_K^{S-PLMC}, \mu_K^{PLMC}) \leq \frac{1}{\sqrt{K}}$. Assuming A, B and C we consider $\eta \leq \min(R^2/K, \alpha/K^2)$, and $K = \tilde{\Omega}(\varepsilon^{-12}d^{12})$. Now using the triangle inequality together with the bounds in Eq.s 7 we have: $TV(\mu_K^{S-PLMC}, \pi) \leq TV(\mu_K^{S-PLMC}, \mu_K^{PLMC}) + TV(\mu^{PLMC}, \pi) \leq \varepsilon + \frac{1}{\sqrt{K}}$.

**S-ProxLMC.** We conclude with a similar argument for $TV(\mu_K^{S-ProxLMC}, \pi)$ using Eq.s 8. Considering $\eta = \min(\gamma(1 + \beta^2\gamma^2)^{-1}, \frac{1}{\delta K^2})$, and $K = \tilde{\Omega}(\varepsilon^{-6}d^5)$, we obtain $(\varepsilon + \frac{1}{\sqrt{K}})$ approximation in TV of the target Gibbs distribution.

$\square$

*Proof of Corollary 1.* **Z-PLMC**: We have:

$$TV(\mu_T^{\tilde{X}}, \mu_T^{\tilde{Y}}) \leq \sqrt{\frac{\eta}{\lambda}\left(\sum_{k=0}^{K-1}\mathbb{E}\left\|G_n U(Y_{k\eta}) - \nabla_x U(Y_{k\eta})\right\|^2 + K\beta^2 R^2\right)}$$

Taking the expectation we have:

$$\mathbb{E}_{g_1 \ldots g_n} TV(\mu_T^{\tilde{X}}, \mu_T^{\tilde{Y}}) \leq \mathbb{E}_{g_1 \ldots g_n}\sqrt{\frac{\eta}{\lambda}\left(\sum_{k=0}^{K-1}\mathbb{E}\left\|G_n U(Y_{k\eta}) - \nabla_x U(Y_{k\eta})\right\|^2 + K\beta^2 R^2\right)}$$

$$\leq \sqrt{\frac{\eta}{\lambda}\left(\sum_{k=0}^{K-1}\mathbb{E}_Y\mathbb{E}_{g_1 \ldots g_n}\left\|G_n U(Y_{k\eta}) - \nabla_x U(Y_{k\eta})\right\|^2 + K\beta^2 R^2\right)} \text{ (Jensen inequality)}$$

Note now that we have:

$$\mathbb{E}_{g_1 \ldots g_n}\left\|G_n U(Y_{k\eta}) - \nabla_x U(Y_{k\eta})\right\|^2 \leq \delta, \forall Y_{k\eta}.$$

For $n \geq \left(\beta\nu(d+2)^{3/2} + (d+1)^{\frac{1}{2}}L\right)^2/\delta$ The rest of the proof is an application of Theorem 2.

**Z-ProxLMC.** A similar argument holds.

$\square$

