# OpenReview forum: "Surrogate-Based Constrained Langevin Sampling With Applications to Optimal Material Configuration Design"
_ICLR.cc/2020/Conference — Reject_

### Official Review · AnonReviewer2 · 2019-10-17
**Official Blind Review #2**

**Rating:** 3

**Review:**

This paper proposes a solution to overcome the challenges due to the black-box nature of physical constraints that are involved in the design of nano-porous templates with optimal thermoelectric efficiency.

Unfortunately, I cannot comment on the overall scientific contribution of the paper, as I do not possess the expertise to judge it accurately. My expertise is so outside of this field that I will rely on the judgement of the other reviewers, whom I hope will have more experience and will better know the literature.

I can only report that the proposed method does not seem to be a particularly good approximation to the zeroth-order method since the mean values for kappa and sigma in table 2 are quite a bit worse than those obtained with the baseline. Of course, the proposed approach is quite a bit faster. However, the paper does not provide a sense of whether these values are actually useful. In practice, would one want to wait longer to get a better quality result, or are the numbers obtained with the proposed approach usable?

Also, could the proposed approach be applied to other problems? It would be great to see at least one or two other areas where this could be applied to, since I doubt the general ICLR audience is well-versed in nano-porous templates.

**Experience Assessment:**

I do not know much about this area.

**Review Assessment: Checking Correctness Of Derivations And Theory:**

I did not assess the derivations or theory.

**Review Assessment: Checking Correctness Of Experiments:**

I did not assess the experiments.

**Review Assessment: Thoroughness In Paper Reading:**

I made a quick assessment of this paper.

---

> ### Author Response · Authors · 2019-11-11
> **Thanks for your feedback, we answered your concerns**
>
> Thank you for your comments.
>
> Q1: “I can only report that the proposed method does not seem to be a particularly good approximation to the zeroth-order method since the mean values for kappa and sigma in table 2 are quite a bit worse than those obtained with the baseline. Of course, the proposed approach is quite a bit faster. However, the paper does not provide a sense of whether these values are actually useful. In practice, would one want to wait longer to get a better quality result, or are the numbers obtained with the proposed approach usable?
>
> We emphasize that zero-order methods are not baselines from previously published work but are our proposed methods. It is also important to note that zero-order and Taylor 1 (Deep Surrogate Model) are complementary, meaning they can work together in a hybrid fashion. We show in the appendix such results where we alternate between using zero-th order and Taylor-1 depending on the descent trajectory. See Algorithm A, Table 3 and Figure 4 for the details. This hybrid strategy provides the best computational time /accuracy tradeoff.
>
> It is worth mentioning that even without the hybrid strategy, configurations produced by our approach including Taylor-1 are practically usable. We consider for this purpose as a baseline configuration with no pores (kappa_reference=1 W/m/K), Taylor-1 produces configurations that give roughly 20-fold reduction of thermal conductivity (kappa/kappa_reference=0.05) with respect to the case with no pores, while controlling the mechanical stability. This may translate into promising nanostructured thermoelectric materials  in practice (See Mildred S. Dresselhaus et al 2007).
>
> New Directions for Low‐Dimensional Thermoelectric Materials. Mildred S. Dresselhaus et al, 2007.
>
>
> Q2: “Also, could the proposed approach be applied to other problems? It would be great to see at least one or two other areas where this could be applied to, since I doubt the general ICLR audience is well-versed in nano-porous templates.”
>
> Please see our general response for the possible applications of our framework.

---

### Official Review · AnonReviewer1 · 2019-10-27
**Official Blind Review #1**

**Rating:** 6

**Review:**

The paper considers the problem of sampling points from a constrained set in R^d where the constraints can only be accessed in a zero order fashion. They consider the specific situation where the constraints are a solution of a complicated PDE solver and hence the derivatives or specific functional forms of the constraint cannot be obtained. They repose the problem as sampling from a Gibbs distribution whose potential contains constraints as penalties in a Lagrangian fashion. They now wish to sample from the Gibbs distribution using Langevin diffusion.  The Langevin process requires a derivative of the gradient. The setting does not allow for that and therefore the authors propose two approaches -

1. Constructing the gradient from zeroth order entries of a gaussian smoothed potential (much like works of Nesterov et al on zero order optimization).
2. Using a parameteric function class (like an RKHS or a neural network) to learn a function which well approximates the gradient of the constraints as well given zeroth order constraint evaluations.

For the latter, the authors propose two approaches. Hermite learning which directly approximates the error on the gradient as well as the function evaluation. However this involves a separate estimate of the gradient (via a zero order approach such as Gaussian smoothing). An alternative which seems more sample efficient is to do direct learning from zero order samples by penalizing first-order Taylor approximation between given points.

The authors provide a theoretical analysis of the all the approaches and sub-approaches given above and further provide experiments to evaluate this on a problem from material design. I am not at all familiar with the domain of the experiments to comment on competing approaches. As far as I can see the authors also compare only their own proposed algorithms which are many. I will focus on the theoretical analysis which seems to form the bulk of the paper anyway.

The theoretical analysis seems quite rigorous as it begins by first providing a basic guarantee for constrained langevin sampling when gradients are computed with error. The non error gradient part of this analysis has been established before and the authors mention the references appropriately. I have a couple of questions regarding the precise statements of the theorem that i will ask towards the end of the review. Overall its hard to comment on the tightness of the analysis as the non-error versions are also unclear of the tightness of the bounds. Nevertheless the bounds achieved do not look much worse than the non-error counterparts and are easy to implement. The rest of the bounds focus on achieving low error in approximation of the gradients in various settings. Overall the theory in this part seems very loose in terms of bounds as exponential factors in dimensions start to appear and in that regard seems quite preliminary but its hard to comment on whether its natural or can be improved.

Overall the paper is a rigorous treatment of multiple components that would go into the problem of sampling zero order constrained sets and merges many ideas which can all be useful. Nevertheless the paper is a little lacking of novelty in the sense that it brings together many existing ideas and provides an analysis of the effect of bringing them together but none of the theory significantly improves over the existing theory.

Some questions I have regarding the theorem statements

Regarding theorem 1 SPLMC convergence  (and corollaries of the theorem) -  I find it surprising that there is no lower bound assumption on eta in terms of K - only an upper bound. This seems wrong particularly as the theorem as stated then allows eta to be set extremely small while K is finite, in which case there should be no convergence theorems at all. The condition on eta should be a theta(f (K)) for some f type of condition like in the second part of the theorem. I would suggest the authors to look into the theorem - or claify why this is the case.

I am confused by the presentation of the Shi et al results as there is no penalty appearing for the approximation error due to an RKHS, only a finite sample penalty. Does the result assume that phi belongs to the function class of the RKHS in question? Probably yes and in that case that should be specified.


**Experience Assessment:**

I have read many papers in this area.

**Review Assessment: Checking Correctness Of Derivations And Theory:**

I assessed the sensibility of the derivations and theory.

**Review Assessment: Checking Correctness Of Experiments:**

I did not assess the experiments.

**Review Assessment: Thoroughness In Paper Reading:**

I read the paper thoroughly.

---

> ### Author Response · Authors · 2019-11-11
> **Thank you for your review, we updated the paper**
>
> Thank you for your detailed review and careful reading.
>
> Q1: “As far as I can see the authors also compare only their own proposed algorithms which are many.”
>
> For sampling in the black-box setting under constraints, we are not aware of any existing method. It should be mentioned that the work of Shen et al. (2019) deals with the unconstrained case that can not be directly applied to our problem of interest.
>
> Q2: “Overall it’s hard to comment on the tightness of the analysis as the non-error versions are also unclear of the tightness of the bounds... Overall the theory in this part seems very loose in terms of bounds as exponential factors in dimensions start to appear and in that regard seems quite preliminary but it’s hard to comment on whether its natural or can be improved.”
>
> Please note that in our bounds we inherit this dependency on dimension from the white box sampling as shown in Bubek et al and in Brosse and Moulines. It is hard to improve on those bounds in the black box setting, since our gradient free methods or the surrogate models are approximations of the white box setting.
>
> Q3: “Nevertheless the paper is a little lacking of novelty in the sense that it brings together many existing ideas and provides an analysis of the effect of bringing them together but none of the theory significantly improves over the existing theory.”
>
> We clarify here that our main innovation in this paper is (i) formulating the constraint satisfaction problem as a sampling from a Boltzmann distribution; (ii) proposing constrained gradient free methods (zero-order) and surrogate models for the black-box sampling problem. We analyzed our proposed methods. Even though we don’t improve on the existing theory in the white box sampling (when the potential of the Boltzmann distribution is known), we present novel methods backed with a theory for the black-box sampling case that was not studied before in the constrained setting. Black-box sampling is of practical interest in many situations such as  inverse design that we consider or adversarial robustness. See also general comment for other applications.
>
> Q4: “Regarding theorem 1 SPLMC convergence  (and corollaries of the theorem) -  I find it surprising that there is no lower bound assumption on eta in terms of K - only an upper bound. This seems wrong particularly as the theorem as stated then allows eta to be set extremely small while K is finite, in which case there should be no convergence theorems at all. The condition on eta should be a theta(f (K)) for some f type of condition like in the second part of the theorem. I would suggest the authors to look into the theorem - or clarify why this is the case.”
>
> Great catch. We agree that the step size \eta should have a lower bound. More precisely, in part 1 of Theorem 2 (S-PLMC), from Bubeck et al. \eta in \Theta(R^2/K), the step size must be lower bounded by R^2/K up to some constant. We have fixed this in the manuscript. Thanks!
>
>
> Q5: “I am confused by the presentation of Shi et al results as there is no penalty appearing for the approximation error due to an RKHS, only a finite sample penalty. Does the result assume that phi belongs to the function class of the RKHS in question? Probably yes and in that case that should be specified.”
>
> Indeed, in Shi et al., the assumption is on the universality of the kernel and that the true function belongs to the RKHS and the approximation error is zero. We will clarify this point in the revised manuscript. Thank you for pointing that.

---

### Official Review · AnonReviewer3 · 2019-10-30
**Official Blind Review #3**

**Rating:** 6

**Review:**

The paper presents novel Langevin sampling implementations with (i) zero order and (ii) deep neural network gradient approximations for constraint satisfaction problems. These approximations improve the computational efficiency by a factor of 17  compared to related black box sampling approaches (Shen et al.).
As a non-domain expert I cannot evaluate the importance of the contributions or the complexity of the tasks. However, given the well written and clearly structured presentation, the theoretical proofs and the generality of the constraint solver (with learned approximated gradients), I would vote for accepting the paper.

Open questions:
- How does the approach relate to "multi-objective Bayesian optimization". The result of this multi objective optimization problem can be also used to further investigate different configurations in different tasks  (generalization via pareto front sampling)?

- The authors note that their approach is closely related to Shen et al. (2019). However, there is no algorithmic comparison in the experiments. Can the work of Shen et al. (2019) also be applied to the constraint setting?

- How crucial or restricting is the assumption of log-concave and smooth target distributions? Which constraint satisfaction problems fall into this class? Can you provide some examples?

- The authors use for approximating the gradients a neural network with 4 hidden layers (with 128, 72, 64 and 32 ReLU neurons) and fine tune the networks in a grid search manny. Which parameters were fine tuned? What were the input and output dimensions. I assume 10 kappa and 10 sigma values were the predicted outputs in individual networks.

Minor Issues:
- Some paragraph titles to no end with a dot, e.g., "Optimization approaches" in page 8
- The last sentence in page 8 is incomplete.
- The computational speed up is a factor of 17 (page 9) or 15 (page 10)?
- Typo in page 13: after the target output is properly normalization

**Experience Assessment:**

I do not know much about this area.

**Review Assessment: Checking Correctness Of Derivations And Theory:**

I assessed the sensibility of the derivations and theory.

**Review Assessment: Checking Correctness Of Experiments:**

I assessed the sensibility of the experiments.

**Review Assessment: Thoroughness In Paper Reading:**

I read the paper at least twice and used my best judgement in assessing the paper.

---

> ### Author Response · Authors · 2019-11-11
> **Thanks for your questions, we added clarifications to the manuscript**
>
> Thank you for your insightful questions.
>
> Q1: “How does the approach relate to "multi-objective Bayesian optimization". The result of this multi objective optimization problem can be also used to further investigate different configurations in different tasks  (generalization via pareto front sampling)?”
>
> Indeed, multi-objective Bayesian optimization is related to our approach and we discussed this in our paper (See Remark 1). However, while Bayesian optimization can produce one single optimal configuration at a time, our method produces a set of candidates. In most applications after the optimal configuration(s) is produced, it is tested in silico or in wet lab to assess its optimality. Sampling approach allows having a ranked set of optimal configurations, which gives the designer the choice to choose among the ones that work the best in reality.
>
> Q2: “The authors note that their approach is closely related to Shen et al. (2019). However, there is no algorithmic comparison in the experiments. Can the work of Shen et al. (2019) also be applied to the constraint setting?”
>
> Shen et al. 2019 approach is not applicable to the constrained setting. The need of the projection step or the proximal step is crucial for the constrained setting, which is not used in Shen et al 2019. Indeed, zero-order projected and proximal Langevin are one of the main contributions of our paper.
>
> Q3: “How crucial or restricting is the assumption of log-concave and smooth target distributions? Which constraint satisfaction problems fall into this class? Can you provide some examples?”
>
> The log-concave and smoothness assumptions are  standard and  crucial for the  analysis of Langevin sampling (see Bubek et al, Moulines et al). Note also that these are standard assumptions in convex optimization. Showing that the zero order methods and surrogate models converge in this setting is a first sanity check for the validity of our approach, and this was assumed also for example in Shen et al 2019 for the unconstrained setting.
>
> In our nano-porous design application, both constraints, which are solutions to PDE solvers, are neither smooth nor log-concave. Raginsky et al extended  the analysis to the non convex case for the unconstrained Langevin. Extending their analysis to the constrained case is non-trivial and is beyond the scope of our paper, but is indeed an important direction for future work.
>
> Non-convex Learning via stochastic gradient Langevin dynamics: a non asymptotic analysis. Maxim Raginsky, Sasha Rakhlin, Matus Telgarsky.
>
>
> Q4: “The authors use for approximating the gradients a neural network with 4 hidden layers (with 128, 72, 64 and 32 ReLU neurons) and fine tune the networks in a grid search manny. Which parameters were fine tuned? What were the input and output dimensions. I assume 10 kappa and 10 sigma values were the predicted outputs in individual networks.”
>
> We mainly fine-tuned the learning rate and decay. The fine-tuning description was hidden by Figure 1 at the end of page 8 in our previous version.
>
> The input dimension is 110, including 20 (x, y)-coordinates of 10 pores and 90 pairwise pore-pore distances along x-axis and y-axis. For each input pattern, we use one network to estimate electrical conductivity value and the other network for mechanical conductivity. Hence, the output is a scalar.
>
> “Minor Issues”
> Thank you for your careful reading. We fixed the typos.

---

### Author Response · Authors · 2019-11-11
**General Response**

We thank the reviewers for the constructive feedback. We provide a general response and address each reviewer separately point by point below.

We motivate the problem from the physical design standpoint because the black-box nature is appealing in the context in which constraints are solutions to complex and expensive PDE solvers. However, our framework is generally applicable to other tasks. We give two such examples: (i) in the black-blox adversarial learning, the gradient of the classifier is not available while the number of evaluations/queries is limited. Our approach can be used to generate a diverse set of adversarial examples (assuming we have samples to train the surrogate model) (see Srinivasan et al 2019); (ii) sampling under black box knowledge constraints where the likelihood and its gradient are not available is ubiquitous beyond the inverse design problem we consider. See for example learning under constraints (Hu et al 2018).

We note that both zero-order and surrogate deep models for Langevin sampling are our own contributions in the constrained setting. Apart from the application side, we make a concrete theoretical contribution by corroborating our approach with convergence analysis.

We made the following updates to our manuscript following the reviewers’ suggestions:

1. Added the lower bound to the step size in Theorem 2 for S-PMLC as pointed by the Reviewer 1 and its Corollary.

2. Added a hybrid method using zero-order and Taylor-1 surrogate model, that alternates between gradient-free and gradient of the surrogate depending on the decrease in the potential function (Objective). This hybrid strategy offers the best trade-off in accuracy and computation. The details can be found in Appendix A with results in Table 3 and Figure 4.

3. We fixed minor typos.

The updates in the manuscripts are highlighted in red.

References:

Robustifying models against adversarial attacks by Langevin dynamics. Vignesh Srinivasan et al, 2019.

Deep generative models via learnable knowledge constraints. Zhiting Hu et al, 2018.

---

### Decision · Program_Chairs · 2019-12-19

**Decision:**

Reject

**Comment:**

The paper is not overly well written and motivated. A guiding thread through the paper is often missing. Comparisons with constrained BO methods would have improved the paper as well as a more explicit link to multi-objective BO. It could have been interesting to evaluate the sensitivity w.r.t. the number of samples in the Monte Carlo estimate. What happens if the observations of the function are noisy? Is there a natural way to deal with this?
Given that the paper is 10+ pages long, we expect a higher quality than an 8-pages paper (reviewing and submission guidelines).